# Association of intraabdominal fat with the risk of incident chronic kidney disease according to body mass index among Korean adults

Jeonghwan Lee[1]◉, Seran Min[2]◉, Seung-Won Oh[3‡]*, Sohee Oh[4], Yoon-Hye Lee[2], Hyuktae Kwon[2], Cheol Min Lee[3], Ho-Chun Choi[3], Nam Ju Heo[5‡]*

1 Department of Internal Medicine, Boramae Medical Center, Seoul National University Hospital, Seoul, South Korea, 2 Department of Family Medicine, Seoul National University Hospital, Seoul National University College of Medicine, Seoul, South Korea, 3 Department of Family Medicine, Healthcare System Gangnam Center of Seoul National University Hospital, Seoul, South Korea, 4 Medical Research Collaborating Center, Seoul National University Boramae Medical Center, Seoul, South Korea, 5 Department of Internal Medicine, Healthcare System Gangnam Center, Seoul National University Hospital, Seoul, South Korea

◉ These authors contributed equally to this work.
‡ SWO and NJH also equally contributed to this work as corresponding authors.
* njheo@snuh.org (NJH); sw.oh@snu.ac.kr (SWO)

**Data Availability Statement:** Dr Oh and Dr Heo had full access to all of the data in the study and take responsibility for the integrity of the data and the accuracy of the data analysis. Data cannot be

## Abstract

### Background

The association between abdominal visceral adipose tissue and the risk of incident chronic kidney disease according to body mass index in the Asian population, remains unclear. We evaluated the impact of abdominal adiposity stratified by body mass index on the risk of incident chronic kidney disease.

### Methods

A cohort study included 11,050 adult participants who underwent health check-ups and re-evaluated the follow-up medical examination at a single university-affiliated healthcare center. Cross-sectional abdominal adipose tissue areas were measured using computed tomography. The primary outcome was progression to chronic kidney disease (estimated glomerular filtration rate <60 ml/min/1.73m$^2$). The highest quartile of visceral adipose tissue was used for the cut-off of central obesity.

### Results

During the mean of 5.6 follow-up years, 104 incident chronic kidney disease cases were identified. The risk for chronic kidney disease incidence was significantly increased in the 3rd and 4th quartile ranges of visceral adipose tissue [hazard ratio (95% confidence interval)]: 4.59 (1.48–14.30) and 7.50 (2.33–24.20), respectively. In the analysis stratified by body mass index, the chronic kidney disease incidence risk was increased in the highest quartile range of visceral adipose tissue in the normal weight group: 7.06 (1.35–37.04). However, there was no significant relationship between visceral adipose tissue and chronic kidney disease in the obese group. Compared to the subjects with normal weight and absent central

shared publicly because of private and sensitive patient information of the participants. The data used in the study are managed in accordance with Seoul National University Hospital IRB regulations. Data are available from the Healthcare System Gangnam Center Healthcare Research Institute Data Access / Ethics Committee (contact via +82-2-2112-5631) or the corresponding authors (njheo@snuh.org; sw.oh@snu.ac.kr) for researchers who meet the criteria for access to confidential data.

**Funding:** This work was supported by a public clinical research grant-in-aid from the Seoul Metropolitan Government Seoul National University (SMG-SNU) Boramae Medical Center (04-2020-3). The funders had no role in study design, data collection and analysis, decision to publish, or preparation of the manuscript.

**Competing interests:** The authors have declared that no competing interests exist.

obesity, the hazard ratio for chronic kidney disease incidence was 2.32 (1.26–4.27) among subjects with normal weight and central obesity and 1.81 (1.03–3.15) among subjects with obesity and central obesity.

## Conclusion

Visceral adipose tissue was a significant risk factor for subsequent chronic kidney disease progression, and the association was identified only in the normal weight group. Normal-weight central obesity was associated with excess risk of chronic kidney disease, similar to the risk in the group with obesity and central obesity.

## Introduction

Chronic kidney disease (CKD) is a global health threat with increasing mortality and socioeconomic expenditures [1–3]. The global prevalence of CKD is estimated at 11–13%; the number of deaths reached 1.2 million in 2015, that had increased by more than 32% since 2005 [4]. CKD prevalence is increasing in several countries due to the increased aging population, increased prevalence of diabetes and hypertension, increased exposure to drugs and environmental chemicals, and lifestyle changes [5]. Obesity is also an important risk factor for CKD and progression to end-stage kidney disease [6, 7].

Body mass index (BMI) is the most commonly used index of obesity; however, it does not take into account body fat distribution. The importance of central obesity for various clinical outcomes, characterized by high abdominal fat distribution, is well known. Central obesity is an independent risk factor for overall mortality, cardiovascular disease, and metabolic syndrome even after adjusting BMI [8–10]. Waist circumference (WC) and waist-to-hip ratio, which reflect visceral adipose tissue (VAT), are usually used for central obesity measurement. Recently, researchers found differential effects between VAT and subcutaneous abdominal adipose tissue (SAT) in metabolic and cardiovascular disease [11–13]. However, methods using WC cannot distinguish between VAT and SAT. Computed tomography (CT) at the level of the lumbar spine is a more accurate method for the assessment of VAT and SAT than methods using WC.

Even among normal-weight individuals, those with central obesity may be at increased risk of mortality. The mortality risk of those with normal-weight central obesity is reportedly similar to or greater than that of overweight or obese patients with central obesity [14, 15]. Central obesity is reportedly associated with increased urinary albumin-to-creatinine ratio or lower estimated glomerular filtration rate (eGFR) [16, 17]. VAT is also independently associated with albuminuria, decreased eGFR, and prevalence of CKD [18–21]. However, bioelectrical impedance analysis, not CT, was used in most previous studies. Additionally, studies regarding normal-weight central obesity defined by VAT with CKD are scarce. Therefore, we aimed to evaluate the effects of abdominal fat measured by CT on incident CKD according to BMI in Korean adults.

## Methods

### Study participants

We enrolled adult participants who underwent routine comprehensive health check-up programs with abdominal CT scans at Seoul National University Hospital Healthcare System

Gangnam Center from January 2005 through December 2010 and underwent at least one more health check-up after 1 year until December 2016. All participants were required to complete a medical questionnaire, including medical history, smoking, exercise, alcohol consumption, and drug history. Among 43,049 participants, 27,441 participants who underwent abdominal CT during the 6-year period were screened (S1 Fig in S1 File). We excluded participants aged <18 years and those with comorbidities such as malignancy, coronary heart disease with percutaneous coronary stent insertion, stroke, hyperthyroidism, and hypothyroidism. Among the 25,105 eligible participants, 11,198 (44.6%) underwent a follow-up health examination. Participants with a short follow-up period below 1 year (n = 114) and low baseline eGFR (<60 ml/min/1.73m$^2$, n = 35) were also excluded. The study protocol adhered to the principles of the Declaration of Helsinki and was approved by the Institutional Review Board of Seoul National University Hospital (1707-151-872). All participants voluntarily underwent health check-ups and paid their own expenses. All of the patient's personal information were encrypted, and the personal number and encrypted code were kept separate from the clinical information. Written informed consent was waived due to the retrospective study design and no infringement on patient privacy or health status.

### Clinical and laboratory assessments

Medical information of participants was retrospectively retrieved from the electronic medical record database. Height, weight, waist circumference, and blood pressure were measured directly by trained nurses. BMI was calculated as the weight in kilograms divided by the square of the height in meters (kg/m$^2$). Blood samples were drawn in the morning after participants had fasted for at least 12 hours to measure serum glucose, triglyceride, LDL, HDL, and total cholesterol levels. Diabetes mellitus was defined as a history of diabetes mellitus, current anti-diabetes medications, fasting glucose ≥126 mg/dl, or HbA1c ≥6.5%. Hypertension was defined as a medical history of hypertension, current anti-hypertensive medications, or measured systolic or diastolic blood pressure ≥140/90 mmHg more than twice at the health check-up. Regular exercise was defined as moderate-intensity exercise for >20 minutes, more than thrice a week. Dietary habits were assessed using structured questionnaires. Salt preference scores were calculated from a sum of three categories: preference for salted soup with a meal; having salted food (fish or vegetables) two or more times a week; and the habit of adding extra salt or soy sauce to the meal. Soda preference was defined as drinking different types of soda (Coke, Sprite, or other sweetened beverages) two or more times a week. Detailed questionnaire according to salt and soda preference and scoring systems are listed in the Supplements. Serum creatinine levels were measured using the Jaffe rate method (kinetic alkaline picrate) calibrated with the isotope dilution mass spectrometry reference method. GFR was calculated using the Chronic Kidney Disease Epidemiology Collaboration (CKD-EPI) equations [21]. CKD was defined as an eGFR below 60 ml/min/1.73m$^2$. When the participants underwent multiple (three times or more) health check-ups, we utilized the second health check-up data in the follow-up study.

### Measurement of VAT and SAT by CT scan

The abdominal adipose tissue areas were measured at the level of the umbilicus using a 16-detector row CT scanner (Somatom Sensation 16, Siemens Medical Solutions, Forchheim, Germany), as previously described [22]. In brief, a 5-mm-thick umbilical-level abdominal section was obtained. The cross-sectional area (cm$^2$) of the abdominal adipose tissue was calculated using Rapidia 2.8 CT software (INFINITT, Seoul, Korea). The VAT area was defined as intra-peritoneal fat bound by the parietal peritoneum or transversalis fascia, and the SAT area

was defined as fat areas external to the abdomen and back muscles. The total adipose tissue (TAT) area was calculated based on the summation of VAT and SAT. Because a clear standard for determining the normal abdominal fat amount has not been established, we used the lowest quartile as the reference group after subdividing abdominal fat amounts by quartile. In this study, we defined central obesity as increased VAT above the 75th percentile.

## Statistical analysis

In the descriptive analysis of demographic and clinical characteristics, continuous variables are expressed as mean and standard deviation, and categorical variables are described numerically with a percentage. The risk of CKD progression according to the area of VAT and SAT (categorized into quartiles) was assessed using Kaplan-Meier survival analysis with log-rank test and Cox proportional hazard model analysis. The proportional hazard assumption was examined using scaled Schoenfeld residuals. Covariates in the multivariable analysis were selected based on the clinical implications of CKD development and after exploration of the statistical association between the covariates and baseline eGFR and CKD progression. The multivariable model used age, sex, baseline eGFR, BMI, hypertension, diabetes mellitus, smoking, dyslipidemia, uric acid, NSAID (non-steroidal anti-inflammatory drug) usage, alcohol consumption, regular exercise, and salt and soda preferences as covariates. The risk of CKD progression stratified with BMI was also evaluated. In the subgroup analysis stratified by BMI, we excluded underweight participants (BMI < 18.5) to compare the risk of VAT on CKD development especially in the normal-weight (18.5 ≤ BMI < 25) group compared to overweight or obese (BMI > 25) group. The risk of standardized VAT (continuous variable with a mean of zero and standard deviation of 1 in each subgroup) on CKD development was calculated and compared between each group using forest plots. In this subgroup analysis, due to the limited number of CKD events in some subgroups (n = 15 in the female group, n = 26 in the diabetes mellitus group, and n = 42 for those under 60 years of age), the multivariable analysis only included age, sex, diabetes mellitus, and BMI to avoid over-adjustments [23]. To compare the diagnostic performances of various obesity indicators for CKD progression, analysis of the receiver operating characteristic (ROC) curve with the Delong test was applied (R package of pROC). Statistical analyses were performed using R (version 4.0.0 for Windows) and IBM SPSS software (version 21.0; Armonk, NY, USA). The significance level was defined as $P$-values <0.05.

## Results

### Characteristics of study subjects

A total of 11,050 adult participants were finally enrolled in this study. Clinical and demographic characteristics are summarized in Table 1. The mean age was 50.3 ± 8.9 years, and 67.4% participants were male. Prevalence of hypertension and diabetes mellitus was 25.7% and 9.1%, respectively. Baseline serum creatinine levels were 0.85 ± 0.14 mg/dl and base eGFR was 95.8 ± 11.0 ml/min/1.73m$^2$. At the time of enrollment, no patients had decreased eGFR (<60 ml/min/1.73m$^2$). Participants were followed up for 5.6 ± 2.6 (range 1.0–11.9) years (S1 Table in S1 File). Follow-up serum creatinine levels were 0.87 ± 0.18 mg/dl and eGFR was 91.9 ± 12.0 ml/min/1.73m$^2$. During the follow-up period, 104 (0.94%) participants progressed to CKD. Overall incidence rate of CKD was 1.69 cases per 1,000 person-years. Progressors to CKD showed higher mean age (61.1 ± 9.0 vs. 50.1 ± 8.8, $P$< 0.001) and male preponderance (85.6% vs. 67.2%, $P$< 0.001) than non-progressors. Prevalence of diabetes mellitus and hypertension was higher among progressors to CKD than among non-progressors. Progressors to CKD showed higher baseline TAT and VAT than non-progressors. However, SAT was comparable between the groups.

**Table 1. Baseline characteristics of study subjects according to the development of chronic kidney disease.**

| | Total participants (N = 11,050) | CKD non-progressors (N = 10,946) | CKD progressors (N = 104) | *P*-value |
|---|---|---|---|---|
| Age, years | 50.3 ± 8.9 (18, 93) | 50.1 ± 8.8 | 61.1 ± 9.0 | < 0.001 |
| Sex, male | 7,446 (67.4%) | 7,357 (67.2%) | 89 (85.6%) | < 0.001 |
| Diabetes mellitus | 1,005 (9.1%) | 979 (8.9%) | 26 (25.0%) | < 0.001 |
| Hypertension | 2,844 (25.7%) | 2,784 (25.4%) | 60 (57.7%) | < 0.001 |
| Systolic blood pressure, mmHg | 116.9 ± 14.6 (62, 186) | 116.8 ± 14.5 | 125.4 ± 15.2 | < 0.001 |
| Diastolic blood pressure, mmHg | 76.2 ± 11.5 (37, 121) | 76.1 ± 11.5 | 80.7 ± 11.3 | < 0.001 |
| Smoking status, (N = 11,030) | | | | 0.806 |
| Never smoker | 4,995 (45.3%) | 4,951 (45.3%) | 44 (42.7%) | |
| Current smoker | 3,741 (33.9%) | 3,703 (33.9%) | 38 (36.9%) | |
| Ex-smoker | 2,294 (20.8%) | 2,273 (20.8%) | 21 (020.4%) | |
| Alcohol drinking | 6,467 (58.5%) | 6,408 (58.5%) | 59 (56.7%) | 0.709 |
| Salt preference score (N = 10,791) | 0.97 ± 1.01 | 0.97 ± 1.01 | 1.08 ± 1.03 | 0.285 |
| Soda preference (N = 10,907) | 1,794 (16.4%) | 1,782 (16.5%) | 12 (11.7%) | 0.187 |
| Regular exercise | 7,147 (64.7%) | 7,071 (64.6%) | 76 (73.1%) | 0.072 |
| NSAID usage | 161 (1.5%) | 159 (1.5%) | 2 (1.9%) | 0.449 |
| Measurements of obesity | | | | |
| Weight, kg | 66.3 ± 11.3 (33.0, 130.1) | 66.3 ± 11.3 | 69.9 ± 9.4 | < 0.001 |
| Body mass index, kg/m$^2$ | 23.8 ± 2.9 (12.8, 40.0) | 23.8 ± 2.9 | 24.7 ± 2.5 | 0.001 |
| Waist circumference, cm | 85.5 ± 7.9 (57.0, 126.5) | 85.4 ± 7.9 | 89.5 ± 6.7 | < 0.001 |
| Areas of abdominal adipose tissue | | | | |
| Total fat, cm$^2$ | 264.6 ± 90.4 | 264.3 ± 90.0 | 302.0 ± 80.3 | < 0.001 |
| Visceral fat, cm$^2$ | 117.4 ± 54.9 | 117.0 ± 54.8 | 158.4 ± 52.3 | < 0.001 |
| Subcutaneous fat, cm$^2$ | 147.2 ± 56.1 | 147.3 ± 56.1 | 143.6 ± 52.8 | 0.506 |
| Visceral vs. subcutaneous fat ratio, % | 85.7 ± 42.3 | 85.4 ± 42.1 | 120.8 ± 49.3 | < 0.001 |
| Other metabolic risk factors | | | | |
| Fasting glucose, mg/dl | 96.9 ± 18.4 | 96.8 ± 18.2 | 107.0 ± 31.6 | 0.001 |
| HbA1c, % (N = 11,002) | 5.8 ± 0.6 | 5.8 ± 0.6 | 6.2 ± 1.1 | 0.001 |
| Uric acid, mg/dl | 5.7 ± 1.4 | 5.7 ± 1.4 | 6.5 ± 1.6 | < 0.001 |
| Cholesterol, total, mg/dl | 195.7 ± 33.4 | 195.7 ± 33.4 | 192.3± 35.1 | 0.294 |
| HDL cholesterol, mg/dl | 53.6 ± 13.1 | 53.7 ± 13.1 | 49.6 ± 10.8 | < 0.001 |
| LDL cholesterol, mg/dl | 119.7 ± 30.4 | 119.7 ± 30.3 | 116.8 ± 33.3 | 0.327 |
| Triglyceride, mg/dl | 112.8 ± 70.2 | 112.6 ± 70.2 | 132.7 ± 72.9 | 0.004 |
| Serum creatinine, mg/dl | 0.85 ± 0.14 | 0.85 ± 0.14 | 1.04 ± 0.14 | < 0.001 |
| Estimated GFR, ml/min/1.73 m$^2$ | 95.8 ± 11.0 | 96.0 ± 10.8 | 75.4 ± 9.9 | < 0.001 |

Results are expressed as frequencies (percentage) and mean values (standard deviation), as appropriate. CKD development was defined as GFR below 60 ml/min/1.73 m$^2$ at follow-up health check-up.

Salt preference score, sum of three categories of salt preferences, i.e., preference of salt added soup at a meal, having salted food (fish or vegetables) two or more times a week, and having processed or instant food two or more times a week.

Soda preference, drinking various types of soda (coke, Sprites, or other sweetened beverages) two or more times a week.

BMI, body mass index; HDL, high-density lipoprotein; LDL, low-density lipoprotein; NSAID, non-steroidal anti-inflammatory drug.

## CKD development according to VAT and SAT

Numbers and probability of CKD progression increased significantly with an increase in VAT (*P* < 0.001, S2 Table in S1 File and Fig 1). The risk of CKD progression increased according to the increase of VAT categorized into quartiles (*P* < 0.001, S2 Table in S1 File). In Kaplan-Meier analysis, the probability of CKD progression increased according to the increase of

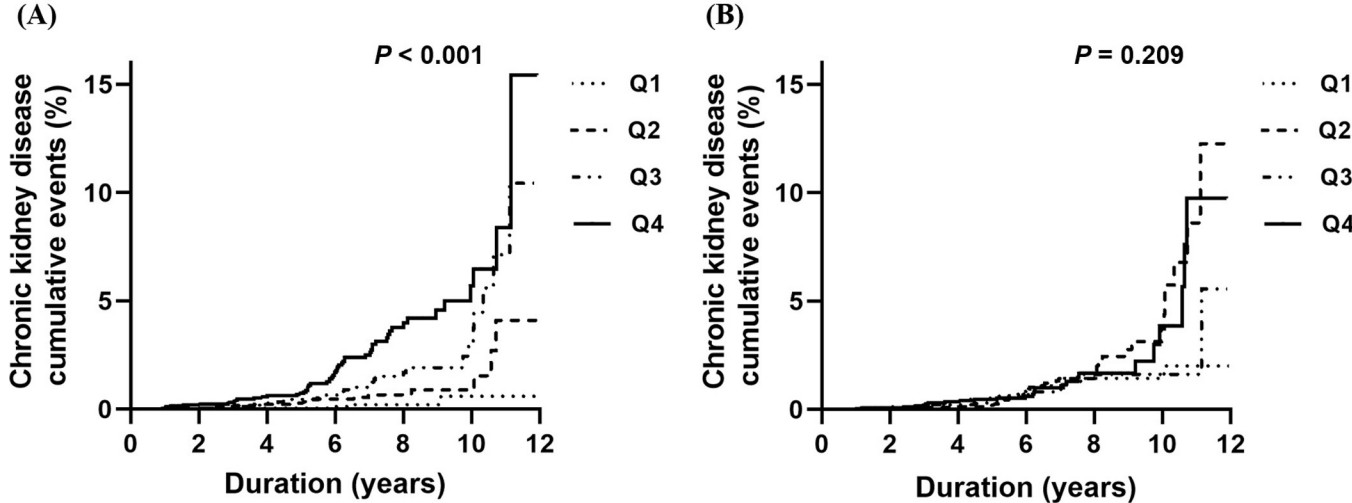

**Fig 1. Probability of progression to chronic kidney disease according to the abdominal adipose tissue area.** (A) Visceral abdominal fat area and risk of progression to chronic kidney disease. (B) Subcutaneous abdominal fat area and risk of progression to chronic kidney disease. Probability of CKD progression increased significantly when the visceral abdominal adipose tissue area was increased ($P < 0.001$, log-rank test). CKD, chronic kidney disease.

VAT ($P < 0.001$, Fig 1). However, an increase in SAT was not associated with CKD progression ($P = 0.284$, S2 Table in S1 File; $P = 0.209$, Fig 1). Table 2 shows the Cox proportional hazard analysis for CKD progression. An increase in VAT was associated with an increased hazard ratio (HR) for CKD development in univariable and multivariable analyses. SAT was not associated with the risk of CKD progression.

We analyzed the effects of VAT on CKD stratified by BMI (Table 3). Underweight populations (BMI < 18.5, n = 317) among the whole participants (n = 11,050) were excluded in this analysis. In the normal-weight group (18.5≤ BMI <25 kg/m$^2$), the risk of CKD development increased with increasing VAT in both univariable and multivariable models. However, in the obese group (BMI ≥25 kg/m$^2$), VAT and incident CKD were not significantly associated. The

**Table 2. The risk of chronic kidney disease in relation to the abdominal adipose tissue area.**

| | | Univariable | | | Multivariable 1 | | | Multivariable 2 | |
|---|---|---|---|---|---|---|---|---|---|
| | N | HR (95% CI) | P-value | N | HR (95% CI) | P-value | N | HR (95% CI) | P-value |
| Visceral abdominal fat area (cm$^2$) | | | < 0.001 | | | 0.001 | | | 0.002 |
| Q1, reference | 2,762 | 1 | | 2,759 | 1 | | 2,676 | 1 | |
| Q2 | 2,763 | 3.63 (1.20–10.93) | 0.022 | 2,757 | 2.80 (0.89–8.79) | 0.078 | 2,654 | 2.76 (0.87–8.77) | 0.085 |
| Q3 | 2,763 | 7.32 (2.57–20.82) | < 0.001 | 2,759 | 4.21 (1.37–12.97) | 0.012 | 2,663 | 4.59 (1.48–14.30) | 0.009 |
| Q4 | 2,762 | 14.85 (5.38–40.94) | < 0.001 | 2,755 | 7.53 (2.37–23.90) | 0.001 | 2,677 | 7.50 (2.33–24.20) | 0.001 |
| Subcutaneous abdominal fat area (cm$^2$) | | | 0.215 | | | 0.688 | | | 0.893 |
| Q$_1$, reference | 2,762 | 1 | | 2,761 | 1 | | 2,657 | | |
| Q$_2$ | 2,763 | 1.57 (0.92–2.66) | 0.097 | 2,760 | 1.13 (0.64–2.01) | 0.677 | 2,671 | 1.10 (0.60–1.97) | 0.776 |
| Q$_3$ | 2,763 | 0.96 (0.53–1.74) | 0.904 | 2,760 | 0.98 (0.50–1.91) | 0.949 | 2,677 | 0.98 (0.49–1.94) | 0.949 |
| Q$_4$ | 2,762 | 1.36 (0.78–2.39) | 0.280 | 2,749 | 0.78 (0.37–1.70) | 0.513 | 2,665 | 0.85 (0.40–1.84) | 0.679 |

N, number of participants; HR, hazard ratio; CI, confidence interval

Q$_1$~Q$_4$, quartile group of each abdominal adipose tissue area

Multivariate model 1 analysis was adjusted for age, sex, baseline eGFR, body mass index, hypertension, diabetes mellitus, smoking, dyslipidemia, and uric acid.

Multivariate model 2 analysis was adjusted for covariates in model 1 and NSAID usage, alcohol consumption, regular exercise, and salt and soda preferences.

**Table 3. The risk of chronic kidney disease in relation to the visceral adipose tissue area according to body mass index (n = 10,733).**

| | | Univariable | | | Multivariable 1 | | | Multivariable 2 | |
|---|---|---|---|---|---|---|---|---|---|
| | N | HR (95% CI) | P-value | N | HR (95% CI) | P-value | N | HR (95% CI) | P-value |
| 18.5 ≤ BMI < 25 | | | | | | | | | |
| Visceral adipose tissue area (cm$^2$) | | | < 0.001 | | | 0.009 | | | 0.011 |
| Q1, reference | 1,794 | 1 | | 1,793 | 1 | | 1,733 | 1 | |
| Q2 | 1,795 | 2.52 (0.65–9.77) | 0.179 | 1,787 | 2.39 (0.45–12.69) | 0.306 | 1,722 | 2.23 (0.39–12.57) | 0.365 |
| Q3 | 1,795 | 3.32 (0.91–12.05) | 0.069 | 1,794 | 2.55 (0.51–12.76) | 0.254 | 1,733 | 2.57 (0.49–13.57) | 0.265 |
| Q4 | 1,794 | 11.01 (3.37–35.94) | < 0.001 | 1,793 | 7.09 (1.42–35.55) | 0.017 | 1,738 | 7.06 (1.35–37.04) | 0.021 |
| BMI ≥ 25 | | | | | | | | | |
| Visceral adipose tissue area (cm$^2$) | | | 0.079 | | | 0.247 | | | 0.260 |
| Q$_1$, reference | 888 | 1 | | 887 | 1 | | 858 | 1 | |
| Q$_2$ | 889 | 1.75 (0.68–4.53) | 0.246 | 887 | 1.22 (0.45–3.30) | 0.691 | 856 | 1.27 (0.44–3.63) | 0.660 |
| Q$_3$ | 889 | 2.71 (1.13–6.48) | 0.026 | 884 | 2.37 (0.92–6.10) | 0.074 | 856 | 2.45 (0.90–6.68) | 0.081 |
| Q$_4$ | 889 | 2.88 (1.18–7.02) | 0.020 | 888 | 1.63 (0.63–4.25) | 0.314 | 863 | 1.67 (0.59–4.71) | 0.335 |

N, numbers of participants; HR, hazard ratio; CI, confidence interval

Q$_1$~Q$_4$, quartile group of each abdominal adipose tissue area

Underweight populations (BMI < 18.5, n = 317) among the whole participants (n = 11,050) were excluded in this analysis.

Multivariate model 1 analysis was adjusted for age, sex, baseline eGFR, body mass index, hypertension, diabetes mellitus, smoking, dyslipidemia, and uric acid.

Multivariate model 2 analysis was adjusted for covariates in model 1 and NSAID usage, alcohol consumption, regular exercise, and salt and soda preferences.

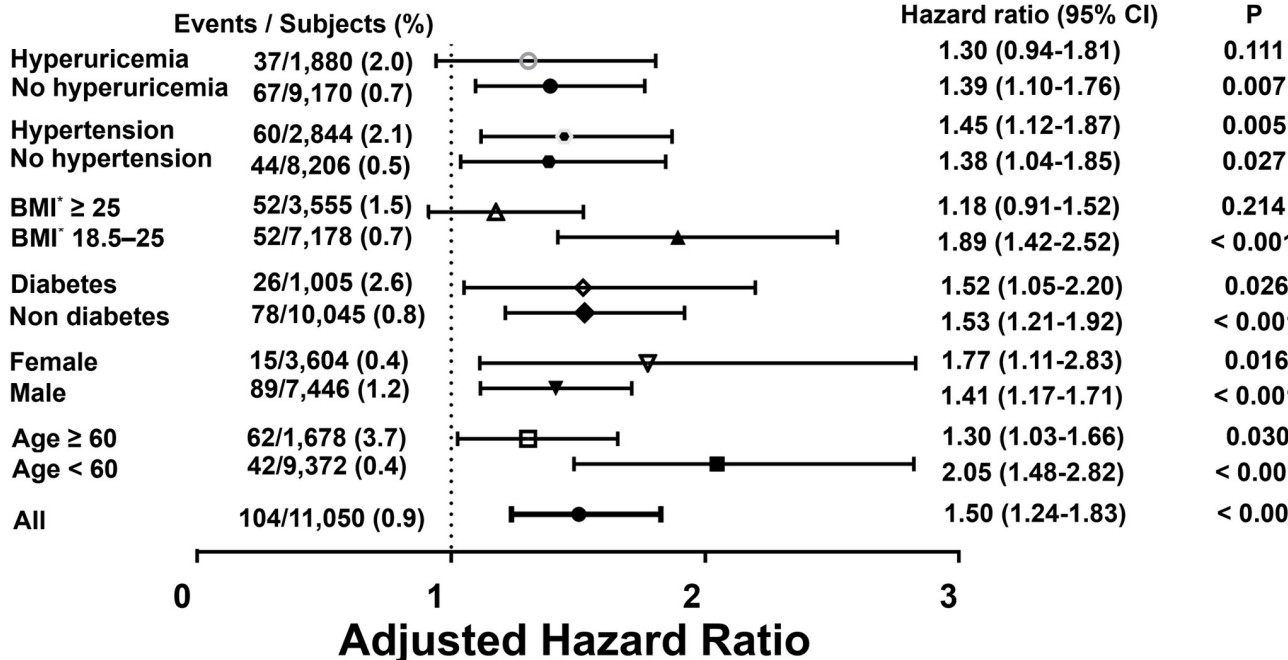

**Fig 2. Forest plot for the risk of chronic kidney disease development due to the increased visceral adipose tissue area.** The risk of CKD development associated with standardized visceral adipose tissue (continuous variables with a mean of zero and standard deviation of 1 in each subgroup) was compared according to age, sex, diabetes mellitus, and BMI. CKD, chronic kidney disease; BMI, body mass index. *In the subgroup analysis stratified by body mass index, underweight populations (BMI < 18.5, n = 317) among the whole participants were excluded.

**Table 4. Hazard ratio for the development of chronic kidney disease based on obesity categorization by body mass index and visceral adipose tissue area (n = 10,733).**

|  | 18.5 ≤ BMI < 25 | 18.5 ≤ BMI < 25 | BMI ≥ 25 | BMI ≥ 25 | P-value |
|---|---|---|---|---|---|
|  | No visceral obesity | Visceral obesity | No visceral obesity | Visceral obesity |  |
| Unadjusted model | 1 (reference) | 5.43 (3.09–9.51) < 0.001 | 2.13 (1.17–3.87) 0.014 | 4.30 (2.63–7.03) < 0.001 | < 0.001 |
| Adjusted mode 1 | 1 (reference) | 2.50 (1.39–4.51), 0.002 | 1.25 (0.67–2.34), 0.490 | 1.97 (1.14–3.41), 0.015 | 0.011 |
| Adjusted mode 2 | 1 (reference) | 2.32 (1.26–4.27), 0.007 | 1.19 (0.63–2.26), 0.597 | 1.81 (1.03–3.15), 0.038 | 0.035 |

BMI, body mass index

Underweight populations (BMI < 18.5, n = 317) among the whole participants (n = 11,050) were excluded in this analysis.

Visceral obesity was defined as the visceral adipose tissue area above 75 percentiles (154.3 cm$^2$).

Model 1 includes covariates of age, sex, diabetes mellitus, baseline eGFR, hypertension, smoking, dyslipidemia, and uric acid.

Model 2 includes covariates of model 2 and NSAID usage, alcohol consumption, regular exercise, and salt and soda preferences.

risk of CKD by VAT in each subgroup was compared (Fig 2 and S3 Table in S1 File). Adjusted HRs for CKD progression was comparable according to diabetes mellitus, hypertension, and hyperuricemia. However, risk for CKD development were more prominent in participants with lower age (<60 years old), female, and normal-weight (18.5 ≤ BMI < 25) group. Sensitivity analysis was performed excluding participants with diabetes mellitus, hypertension, and hyperuricemia (S4 Table in S1 File). Increased VAT was consistently associated with increased risk for incident CKD in all sensitivity analysis model.

S2 Fig and S5 Table in S1 File show the results of receiver operating characteristics for CKD development among various obesity indicators. The area under the curve (AUC) was 0.71, 0.64, and 0.52 for VAT, TAT, and SAT, respectively. SAT [AUC 0.52, 95% confidence interval (CI) 0.47–0.57] was not a useful indicator for CKD development. AUC of VAT was superior to that of body weight, BMI, and WC. The ratio of VAT per SAT showed a similar AUC (0.71, 95%CI 0.67–0.76) compared to VAT itself.

### Association of normal-weight central obesity assessed by VAT with CKD development risk

Association of different combinations of BMI and central obesity assessed by VAT with CKD risk was compared (Table 4). The highest quartile of VAT was used for determining the cut-off of central obesity. The numbers of participants and CKD events with proportions according to obesity defined by BMI and central obesity by VAT were noted in S6 Table in S1 File. Compared to subjects with normal weight and no central obesity, those with obesity and central obesity showed an increased risk for CKD development in the unadjusted and adjusted model (HR 1.81, 95% CI 1.03–3.15). Notably, subjects with normal-weight central obesity had an even higher risk (HR 2.32, 95% CI 1.26–4.27) than those with obesity and central obesity. However, the risk of CKD in subjects with obesity and no central obesity was not significantly higher than that in the reference group with normal weight and no central obesity.

### Discussion

This study demonstrated that VAT is an independent risk factor for incident CKD. SAT was not associated with CKD development. Among various indices for general and central obesity, VAT was superior for predicting incident CKD. In the subgroup analysis by BMI, VAT was associated with CKD risk only in the non-obese group. The group with normal weight and

central obesity had an elevated risk of CKD, similar to the risk in the group with obesity and central obesity.

Obesity is a well-known risk factor for CKD [14]. However, most previous studies were cross-sectional and did not consider the abdominal adipose tissue distribution status. Several previous studies had explored the relationship between obesity and CKD development in a prospective design. A large, longitudinal study of 62,249 participants in Korea found that incident CKD increased in metabolically healthy overweight and obese groups [15]. A meta-analysis showed that obesity with high BMI predicts the development of CKD; however, overweight itself does not [24]. Recently, two studies investigated the impact of abdominal adipose tissue distribution confirmed by CT on the subsequent CKD development, and the results showed some different findings according to the various characteristics of enrolled participants. Madero et al. enrolled 2,489 elderly US participants with a mean age of 74 years old, and the median follow-up duration was 8.9 years with 17% CKD incidence. They revealed that both VAT and SAT were associated with kidney function decline of >30%, and only VAT was significantly associated with incident CKD [25]. Olivo et al. investigated 1,477 African Americans with a majority of females (63%) and followed up for a median 8 (6–11) years. They reported that VAT categorized into quartiles was not associated with incident CKD and the risk for CKD according to VAT (in continuous analysis model) showed U-shaped nonlinear relationship [26]. Effects of VAT on incident CKD had significant interaction with dietary quality and VAT was associated with CKD development only in the low dietary quality group. Discrepancies with our studies' results could be caused by the differences in enrolled participants' ethnicity, gender distribution, dietary quality, and metabolic profiles. The number of enrolled participants in our study was larger than that in other previous studies, and all participants were Asian. The prevalence of diabetes mellitus and hypertension, 9.1% and 25.7%, respectively, was also lower than that in other studies. Our study's participant showed male preponderance. In general, Asian population have low prevalence of obesity and metabolic risk for CKD compared to the African Americans [27]. In addition, sexual difference is an important determinant for obesity-related CKD development [28, 29]. Moreover, it has been reported that there is a difference in food intake pattern and quality between different ethnicities. Asian groups scored highly for diet quality due to higher fish intakes and lower fat intakes compared to other groups, whereas African American groups were noted to be among the least fruit and vegetable eaters [30]. These differences in dietary pattern according to ethnicity can be an important determinant for the CKD development [31, 32]. Madero et al. reported results similar to ours where VAT, but not SAT, was significantly associated with incident CKD [25]. However, they did not analyze the differential impact of VAT on incident CKD according to the general or central obesity status.

Excess body weight and central body fat distribution increase the filtration fraction by increasing the eGFR compared to the effective renal plasma flow and ultimately lead to glomerular hyper-perfusion, hypertension, and kidney failure [17]. Recently, our investigators reported that abdominal adiposity was significantly associated with glomerular hyperfiltration [33]. Moreover, VAT has been recognized as pathogenic fat deposition, and it is more strongly associated with metabolic risk factors than SAT [12]. VAT is a key regulator of numerous adipokines and cytokines including angiotensin, and positively correlated with inflammatory reaction, oxidative stress, impaired endothelial function, and atherosclerosis [34]. In obese people, visceral adipocytes become hypertrophied, and inflammatory reactions become more prevalent. When the inflammatory reaction predominates, insulin resistance increases, leading to oxidative stress, inflammation, fibrosis, and loss of kidney function [35]. Hence, VAT induces glomerulosclerosis and tubulointerstitial fibrosis and ultimately causes CKD [36, 37]. Although we could not investigate the metabolic effect of mediators associated with abdominal

adipose tissue, this explanation can be also applied to the result of this study showing an increased risk for CKD with an increase in VAT.

The interesting findings of this study are that the risk of VAT on CKD is different according to BMI, and the impact of central obesity defined by VAT was prominent in the normal BMI group. The detrimental effects of central obesity especially in normal BMI populations have also been observed in other studies, while the individual results are different. Sahakyan et al. analyzed 15,184 adults in National Health and Nutrition Examination Survey (NHANES) III and reported that central obesity was associated with increased mortality even in normal BMI participants, and normal-weight central obesity men showed the highest mortality risk compared to those with general obesity or central obesity with high BMI [38]. Hamer et al. analyzed a total of 42,702 UK adult participants according to BMI with or without central obesity and mortality [39]. All-cause mortality risk was the highest among normal-weight central obesity subjects [HR 1.22 (1.11–1.35)] compared to overweight with central obesity subjects [HR 1.11 (0.92–1.08)]. These two studies showed similar results wherein central obesity with normal BMI participants had the worst survival outcome. However, a study of 156,624 postmenopausal women in the WHI cohort showed that normal-weight central obesity in women was associated with excess risk of mortality, similar to that of BMI-defined obesity and central obesity [40]. Asian studies are hard to find, but a cross-section study of 117,163 Japanese middle-aged adults showed that normal-weight central obesity was associated with cardiovascular risk factors, such as hypertension, dyslipidemia, and diabetes, and the risk was not higher than that of the group with obesity and central obesity [41].

Most previous studies used WC for evaluating central obesity. Although several studies had explored the association between VAT and GFR decline or CKD development, studies on normal-weight central obesity are scarce. To the best of our knowledge, our study is the first large-scale cohort study investigating the clinical impact of normal-weight central obesity on CKD development risk. There can be several plausible explanations on the mechanisms of normal-weight central obesity and CKD development. Central obesity with normal BMI subjects tends to have relatively low fat-free mass including bone and muscle. Muscle mass is known to be associated with favorable metabolic profiles [39]. Therefore, relatively low muscle mass in normal-weight central obesity subjects might cause poor clinical outcomes due to the loss of protective effects associated with high muscle mass. In addition, normal-weight central obesity subjects might have low gluteofemoral adipose tissue, which is associated with protective metabolic and cardiovascular effects [42]. Another factor associated with this paradoxical and prominently harmful effect of central obesity among non-obese subjects might be the protective effect of high BMI with better nutritional status and low metabolic rates [43].

Our study has several strengths and unique clinical implications. First, we enrolled large-scale middle-aged Asian participants and followed them over 5 years. Considering that Asian populations are relatively non-obese and have lower risk factors for CKD compared with other ethnicities, this study results might be applied to healthier people than those included in previous studies. Second, CT, which can measure VAT and SAT more accurately, was used for evaluating central obesity. In addition, our study is important especially in terms of CKD prevention and public health improvement. In clinical practice, abdominal obesity is easily overlooked in non-obese patients, and interventions to control central obesity are not implemented. This study results with increased CKD development risk in normal BMI population underscore the importance of monitoring for central obesity, besides body weight measurement, and risk reduction interventions for lifestyle modification including a healthy diet and regular exercise, especially in normal body weight patients. However, this study was performed only at a single university-affiliated healthcare center. Therefore, the results of our study need to be validated in a more heterogeneous population with comorbidities or decreased kidney function.

This study found that VAT measured by CT is an independent risk factor for incident CKD in the normal weight group, and subjects with normal-weight central obesity were at higher risk of CKD than those with normal weight and no central obesity. The risk of visceral abdominal obesity tends to be overlooked in normal-weight individuals or Asians, and a public health program that informs the risk of visceral abdominal obesity over obesity itself can help public health improvement. Clinicians need to educate patients with kidney disease about the risks of visceral abdominal obesity, even in non-obese patients. Moreover, discussions on a routine report of VAT measurement in case of abdominal CT examination at health check-ups are needed. Further interventional studies investigating the relationship between efforts of improving central obesity and clinical outcomes including incident CKD, GFR decline, or end-stage kidney disease development in the general and CKD populations are warranted.

## Supporting information

**S1 File.**
(DOCX)

## Acknowledgments

We thank to all the participants in this study.

## Author Contributions

**Conceptualization:** Seung-Won Oh, Hyuktae Kwon, Cheol Min Lee, Ho-Chun Choi, Nam Ju Heo.

**Data curation:** Jeonghwan Lee, Seran Min.

**Formal analysis:** Jeonghwan Lee, Seung-Won Oh, Sohee Oh, Nam Ju Heo.

**Funding acquisition:** Jeonghwan Lee.

**Investigation:** Jeonghwan Lee, Seung-Won Oh, Nam Ju Heo.

**Methodology:** Seung-Won Oh, Yoon-Hye Lee, Hyuktae Kwon, Cheol Min Lee, Ho-Chun Choi, Nam Ju Heo.

**Project administration:** Seung-Won Oh, Nam Ju Heo.

**Resources:** Seung-Won Oh, Nam Ju Heo.

**Software:** Jeonghwan Lee, Seung-Won Oh, Nam Ju Heo.

**Supervision:** Seung-Won Oh, Nam Ju Heo.

**Validation:** Seung-Won Oh.

**Visualization:** Jeonghwan Lee.

**Writing – original draft:** Jeonghwan Lee, Seran Min.

**Writing – review & editing:** Seung-Won Oh, Nam Ju Heo.

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
