## [Decision Letter · Decision Letter 0]

8 Aug 2022

PONE-D-22-12853Association of v isceral adipose tissue with the risk of incident chronic kidney disease according to body mass index among Korean adultsPLOS ONE

Dear Dr. Nam Ju Heo,

Thank you for submitting your manuscript to PLOS ONE. After careful consideration, we feel that it has merit but does not fully meet PLOS ONE’s publication criteria as it currently stands. Therefore, we invite you to submit a revised version of the manuscript that addresses the points raised during the review process.

We look forward to receiving your revised manuscript.

Kind regards,

Paolo Magni

Academic Editor

PLOS ONE

Journal Requirements:

Additional Editor Comments (if provided):

Please address all reviewer's comments.

Reviewers' comments:

Reviewer's Responses to Questions

**Comments to the Author**

1. Is the manuscript technically sound, and do the data support the conclusions?

Reviewer #1: Partly

2. Has the statistical analysis been performed appropriately and rigorously? 

Reviewer #1: Yes

3. Have the authors made all data underlying the findings in their manuscript fully available?

Reviewer #1: Yes

4. Is the manuscript presented in an intelligible fashion and written in standard English?

Reviewer #1: Yes

5. Review Comments to the Author

Reviewer #1: The manuscript addresses the study of CKD compared to the BMI of exclusively Korean individuals using CT. In general, the data showed that individuals with a normal BMI, but with visceral adipose tissue (VAT) had a higher risk of CKD than individuals with a higher BMI associated with a lower muscle mass. Other studies have already shown this contradictory effect in which obese individuals have a lower risk of CKD, however without assessing VAT by means of CT. This study corroborates other findings in this context and reinforces the importance of evaluating the TAV with greater precision, for example, through CT, and thus establishing clinical methods to monitor as well as reduce the formation of TAV.

1 – In the full title, must correct the word visceral

2 – Initials NSAID in the line 172 (Statistical analysis) must be defined

3 – To better understanding - lines 208, 209 “…VAT (P < 0.001, Supplementary Table 1 and Fig. 1A). However, SAT was not associated with 209 CKD progression (Fig. 1B).”

4 - Lines 263-267 the authors report ”Olivo et al. investigated 1,477 African Americans with a majority of females (63%) and followed up for a median 8 (6-11) years with 12.4% CKD incidence. They reported that VAT was not associated with incident CKD [26].” In fact, according Olivo et al “BMI and waist circumference z-score were not associated with incident CKD when modeled continuously (P=0.1 for each). However, higher visceral adipose volume was associated with an increased risk of incident CKD overall (P=0.008), but was nonlinear (Figure 1; P=0.02).” By this reason, the authors must review their discussion at this point.

5 – Lines 270-271 - “In the study of Olivo et al., VAT might be associated with an insignificant risk of incident CKD due to the relatively low numbers of participants [26]”. However, in the study by Olivo et al there is the information "The key strength of our investigation was the use of a large, prospective, high-risk study population with long-term follow-up." Thus, there is a contradiction between the manuscript and Olivo's study, which implies the need to review the discussion by the authors.

6. PLOS authors have the option to publish the peer review history of their article (what does this mean?). If published, this will include your full peer review and any attached files.

Reviewer #1: No

---

## [Author Response · Author response to Decision Letter 0]

16 Sep 2022

Response: On behalf of all coauthors, I would like to thank the editor and anonymous reviewers for the critical comments on this manuscript. We have addressed all comments and revised the manuscript accordingly. Please refer to our changes and detailed explanations regarding the reviewers’ comments in the revised manuscript and rebuttal letter. All changes are shown in blue font both below and in the revised manuscript.

Reviewer Comments:

Reviewer 1

The manuscript addresses the study of CKD compared to the BMI of exclusively Korean individuals using CT. In general, the data showed that individuals with a normal BMI, but with visceral adipose tissue (VAT) had a higher risk of CKD than individuals with a higher BMI associated with a lower muscle mass. Other studies have already shown this contradictory effect in which obese individuals have a lower risk of CKD, however without assessing VAT by means of CT. This study corroborates other findings in this context and reinforces the importance of evaluating the TAV with greater precision, for example, through CT, and thus establishing clinical methods to monitor as well as reduce the formation of TAV. 

Response: We appreciate the reviewer’s valuable comments and positive evaluation of our revisions.

1 – In the full title, must correct the word visceral

Response: We appreciate the reviewer's valuable comments. Following the reviewer’s comment, we fixed the word "visceral adipose tissue" to the intraabdominal fat. 

2 – Initials NSAID in the line 172 (Statistical analysis) must be defined. 

Response: Thank you for valuable comment. We added the have full name of NSAID abbreviation as follows: The multivariable model used age, sex, baseline eGFR, BMI, hypertension, diabetes mellitus, smoking, dyslipidemia, uric acid, NSAID (non-steroidal anti-inflammatory drug) usage, alcohol consumption, regular exercise, and salt and soda preferences as covariates.

3 – To better understanding - lines 208, 209 “…VAT (P < 0.001, Supplementary Table 1 and Fig. 1A). However, SAT was not associated with CKD progression (Fig. 1B).”

Response: We appreciate the reviewer's important comment. According to the reviewer’s comment, the following details were provided regarding existing sentences to help readers better comprehend them: “Numbers and probability of CKD progression increased significantly with an increase in VAT (P < 0.001, S1 Table and Fig 1). The risk of CKD progression increased according to the increase of VAT categorized into quartiles (P < 0.001, S1 Table). In Kaplan-Meier analysis, the probability of CKD progression increased according to the increase of VAT (P < 0.001, Fig. 1). However, an increase in SAT was not associated with CKD progression (P = 0.284, S1 Table; P = 0.209, Fig. 1).”

4 - Lines 263-267 the authors report ”Olivo et al. investigated 1,477 African Americans with a majority of females (63%) and followed up for a median 8 (6-11) years with 12.4% CKD incidence. They reported that VAT was not associated with incident CKD [26].” In fact, according Olivo et al “BMI and waist circumference z-score were not associated with incident CKD when modeled continuously (P=0.1 for each). However, higher visceral adipose volume was associated with an increased risk of incident CKD overall (P=0.008), but was nonlinear (Figure 1; P=0.02).” By this reason, the authors must review their discussion at this point.

Response: Thank you for valuable comment. Olivo et al. summarized and concluded their investigation as “we observed an association between higher visceral adiposity and an increased risk for new-onset CKD in African American adults that was only present among those with poor diet quality”. However, as the reviewer pointed out, we agree that the way of description on previous research of Olive et al. had chance of misunderstanding. Therefore, we added detailed explanations on previous investigation as follows: “VAT categorized into quartiles was not associated with incident CKD and the risk for CKD according to VAT (in continuous analysis model) showed U-shaped nonlinear relationship [26]. Effects of VAT on incident CKD had significant interaction with dietary quality and VAT was associated with CKD development only in the low dietary quality group.”

5 – Lines 270-271 - “In the study of Olivo et al., VAT might be associated with an insignificant risk of incident CKD due to the relatively low numbers of participants [26]”. However, in the study by Olivo et al there is the information "The key strength of our investigation was the use of a large, prospective, high-risk study population with long-term follow-up." Thus, there is a contradiction between the manuscript and Olivo's study, which implies the need to review the discussion by the authors.

Response: We appreciate the reviewer' important comment. Following the comment of the reviewer, we changed the discussion in our manuscript so that it doesn't contradict what Olivo et al. described in their study. “Discrepancies with our studies’ results could be caused by the differences in enrolled participants’ ethnicity, gender distribution, dietary quality, and metabolic profiles”. “Our study’s participant showed male preponderance. In general, Asian population have low prevalence of obesity and metabolic risk for CKD compared to the African Americans [27]. In addition, sexual difference is an important determinant for obesity-related CKD development [28, 29]. Moreover, it has been reported that there is a difference in food intake pattern and quality between different ethnicities. Asian groups scored highly for diet quality due to higher fish intakes and lower fat intakes compared to other groups, whereas African American groups were noted to be among the least fruit and vegetable eaters [30]. These differences in dietary pattern according to ethnicity can be an important determinant for the CKD development [31, 32]. 

27. Sabanayagam C, Lim SC, Wong TY, Lee J, Shankar A, Tai ES (2010) Ethnic disparities in prevalence and impact of risk factors of chronic kidney disease. Nephrol Dial Transplant 25: 2564-2570.https://doi.org/10.1093/ndt/gfq084

28. Chen IJ, Hsu LT, Lu MC, Chen YJ, Tsou MT, Chen JY (2021) Gender Differences in the Association Between Obesity Indices and Chronic Kidney Disease Among Middle-Aged and Elderly Taiwanese Population: A Community-Based Cross-Sectional Study. Front Endocrinol (Lausanne) 12: 737586.https://doi.org/10.3389/fendo.2021.737586

29. Sakurai M, Kobayashi J, Takeda Y, Nagasawa SY, Yamakawa J, Moriya J, et al. (2016) Sex Differences in Associations Among Obesity, Metabolic Abnormalities, and Chronic Kidney Disease in Japanese Men and Women. J Epidemiol 26: 440-446.https://doi.org/10.2188/jea.JE20150208

30. Bennett G, Bardon LA, Gibney ER (2022) A Comparison of Dietary Patterns and Factors Influencing Food Choice among Ethnic Groups Living in One Locality: A Systematic Review. Nutrients 14.https://doi.org/10.3390/nu14050941

31. Asghari G, Momenan M, Yuzbashian E, Mirmiran P, Azizi F (2018) Dietary pattern and incidence of chronic kidney disease among adults: a population-based study. Nutr Metab (Lond) 15: 88.https://doi.org/10.1186/s12986-018-0322-7

32. Bach KE, Kelly JT, Palmer SC, Khalesi S, Strippoli GFM, Campbell KL (2019) Healthy Dietary Patterns and Incidence of CKD: A Meta-Analysis of Cohort Studies. Clin J Am Soc Nephrol 14: 1441-1449.https://doi.org/10.2215/CJN.00530119

---

## [Decision Letter · Decision Letter 1]

27 Oct 2022

PONE-D-22-12853R1

Association of intraabdominal fat with the risk of incident chronic kidney disease according to body mass index among Korean adults

PLOS ONE

Dear Dr. Heo,

Thank you for submitting your manuscript to PLOS ONE. After careful consideration, we feel that it has merit but does not fully meet PLOS ONE’s publication criteria as it currently stands. Therefore, we invite you to submit a revised version of the manuscript that addresses the points raised during the review process.

We look forward to receiving your revised manuscript.

Kind regards,

Paolo Magni

Academic Editor

PLOS ONE

Additional Editor Comments:

Plesse address all reviewer's comments.

Reviewers' comments:

Reviewer's Responses to Questions

**Comments to the Author**

1. If the authors have adequately addressed your comments raised in a previous round of review and you feel that this manuscript is now acceptable for publication, you may indicate that here to bypass the “Comments to the Author” section, enter your conflict of interest statement in the “Confidential to Editor” section, and submit your "Accept" recommendation.

Reviewer #2: All comments have been addressed

Reviewer #3: (No Response)

2. Is the manuscript technically sound, and do the data support the conclusions?

Reviewer #2: Yes

Reviewer #3: Partly

3. Has the statistical analysis been performed appropriately and rigorously? 

Reviewer #2: Yes

Reviewer #3: Yes

4. Have the authors made all data underlying the findings in their manuscript fully available?

Reviewer #2: Yes

Reviewer #3: Yes

5. Is the manuscript presented in an intelligible fashion and written in standard English?

Reviewer #2: Yes

Reviewer #3: Yes

6. Review Comments to the Author

Reviewer #2: Recently, central obesity affects metabolic syndrome, albuminuria and worse outcomes in the kidney. Present study well demonstrated visceral adiposity is associated with incident CKD in general population. Lee et al evaluated visceral adiposity using abominal fat CT. More accurate evaluation for central obeisty is a strength in this study.

Reviewer #3: The authors evaluated the association of intraabdominal fat (measured by computed tomography) with the risk of chronic kidney disease (CKD) in a large Korean population. This study found that visceral adipose tissue (VAT) instead of subcutaneous abdominal fat was associated with incident CKD, especially among participants with normal weight and central obesity. The message was straightforward.

However, the major concern is that VAT has been a known risk factor for diabetes, hypertension, and hyperuricemia. These three diseases are all major contributors to the development of CKD. It is easy to be concerned that VAT is not an independent risk factor for CKD, although adjustment for diabetes, blood pressure, and uric acid levels. To address this issue, I am curious about whether the authors could do the below analyses if possible.

1. Stratified analyses by diabetes (has been done), hypertension, and hyperuricemia;

2. Sensitivity analysis excluding all the participants with diabetes, hypertension, and hyperuricemia;

3. Mediation analyses to explore the mediation effects of these three diseases on the association between VAT and CKD;

4. Provide the participants’ characteristics shown in table 1 by CKD at the follow-up health examination.

7. PLOS authors have the option to publish the peer review history of their article (what does this mean?). If published, this will include your full peer review and any attached files.

Reviewer #2: No

Reviewer #3: No

---

## [Author Response · Author response to Decision Letter 1]

25 Dec 2022

Response: On behalf of all coauthors, I would like to thank the editor and anonymous reviewers for the critical comments on this manuscript. We have addressed all comments and revised the manuscript accordingly. Please refer to our changes and detailed explanations regarding the reviewers’ comments in the revised manuscript and rebuttal letter. All changes are shown in blue font both below and in the revised manuscript.

Reviewer Comments:

Reviewer #2. 

Recently, central obesity affects metabolic syndrome, albuminuria and worse outcomes in the kidney. Present study well demonstrated visceral adiposity is associated with incident CKD in general population. Lee et al evaluated visceral adiposity using abdominal fat CT. More accurate evaluation for central obesity is a strength in this study.

Response: We appreciate the reviewer’s valuable comments and positive evaluation of our revisions.

Reviewer #3. 

The authors evaluated the association of intraabdominal fat (measured by computed tomography) with the risk of chronic kidney disease (CKD) in a large Korean population. This study found that visceral adipose tissue (VAT) instead of subcutaneous abdominal fat was associated with incident CKD, especially among participants with normal weight and central obesity. The message was straightforward.

However, the major concern is that VAT has been a known risk factor for diabetes, hypertension, and hyperuricemia. These three diseases are all major contributors to the development of CKD. It is easy to be concerned that VAT is not an independent risk factor for CKD, although adjustment for diabetes, blood pressure, and uric acid levels. To address this issue, I am curious about whether the authors could do the below analyses if possible.

1. Stratified analyses by diabetes (has been done), hypertension, and hyperuricemia.

Response: We appreciate the reviewer’s valuable comments and positive evaluation of our revisions. Subgroup analysis results stratified by hypertension and hyperuricemia was added to Fig 2. Risk for incident CKD was prominently discriminated according to age, sex, and body mass index. However, we could not find apparent difference in the risk for incident CKD according to the diabetes mellitus, hypertension, and hyperuricemia in multivariable analysis model. We added the results in the Result section as follows: “The risk of CKD by VAT in each subgroup was compared (Fig 2 and S3 Table). Adjusted HRs for CKD progression was comparable according to diabetes mellitus, hypertension, and hyperuricemia. However, risk for CKD development were more prominent in participants with lower age (<60 years old), female, and normal-weight (18.5 ≤ BMI < 25) group.”

2. Sensitivity analysis excluding all the participants with diabetes, hypertension, and hyperuricemia.

Response: Thank you for valuable comment. To clarify the results of this study, we additionally performed sensitivity analysis excluding all the participants with diabetes, hypertension, and hyperuricemia. The results of this sensitivity analysis are added in S4 Table. VAT was significantly associated with incident CKD in all sensitivity analysis model. We added this result in the Result section as follows: “Sensitivity analysis was performed excluding participants with diabetes mellitus, hypertension, and hyperuricemia (S4 Table). Increased VAT was consistently associated with increased risk for incident CKD in all sensitivity analysis model.”

3. Mediation analyses to explore the mediation effects of these three diseases on the association between VAT and CKD.

Response: We appreciate the reviewer's important comment. Central obesity is a key component of metabolic syndrome and well-known risk factor for insulin-resistance. Consequently, it is plausible that comorbidities such as diabetes, hypertension, and hyperuricemia mediated the effects of VAT on incident CKD. Following the valuable comments of the reviewer, we performed mediation analysis using the R Medflex package software (Steen J, Loeys T, Moerkerke B, Vansteelandt S. Medflex: an R package for flexible mediation analysis using natural effect models. J Stat Softw 2017. doi: https://doi.org/10.18637/jss.v076.i11.) 

Mediation analysis results (refer to the inserted table in the rebuttal letter)

Mediator

Diabetes mellitus

HR (95% CI) Hypertension

HR (95% CI) Hyperuricemia

HR (95% CI) Uric acid

HR (95% CI)

Indirect effect 1.053 

(0.893, 1.233) 1.056 

(0.891, 1.243) 1.028 

(0.869, 1.212) 1.024 

(0.861, 1.210)

Direct effect 2.355 

(2.104, 2.630) 2.434 

(2.166, 2.716) 2.438 

(1.179, 2.744) 2.427 

(2.160, 2.714)

Total effect 2.480 

(2.068, 2.968) 2.572 

(2.142, 3.083) 2.506 

(2.095, 3.021) 2.486 

(2.071, 2.983)

As shown in the table above, none of the indirect effects of diabetes, hypertension, hyperuricemia, and uric acid were confirmed to be significant. However, direct effect of VAT was consistently associated with increased risk for incident CKD in all mediation analysis model. There are several possible explanations for why the mediating effects of diabetes, hypertension, and hyperuricemia were not validated in this study's analytical results. First, a follow-up period of 5.6 years is relatively short. Generally, it is known that diabetes mellitus can lead to CKD in 5 to 10 years or more. In addition, as noted in the discussion section, the enrolled participants are all Asian and the prevalence of diabetes mellitus and hypertension are low compared to the other studies (Madero et al, doi.org/10.2215/CJN.07010716 and Olivo et al. doi.org/10.1093/ndt/gfx230). In this study, multivariate analysis including hypertension, diabetes mellitus, and hyperuricemia was conducted to adjust the effects from these comorbidities. Longitudinal prospective follow-up research with a larger sample size and longer duration is required to confirm the mediation effect of these comorbidities. 

4. Provide the participants’ characteristics shown in table 1 by CKD at the follow-up health examination.

Response: Thank you for valuable comment. We added the descriptive statistics for the study subjects at the time of follow-up in the Supplementary Table 1. We added this result in the Result section as follows: “At the time of enrollment, no patients had decreased eGFR (<60 ml/min/1.73m2). Participants were followed up for 5.6 ± 2.6 (range 1.0-11.9) years (S1 Table). Follow-up serum creatinine levels were 0.87 ± 0.18 mg/dl and eGFR was 91.9 ± 12.0 ml/min/1.73m2. During the follow-up period, 104 (0.94%) participants progressed to CKD. Overall incidence rate of CKD was 1.69 cases per 1,000 person-years. Progressors to CKD showed higher mean age (61.1 ± 9.0 vs. 50.1 ± 8.8, P< 0.001) and male preponderance (85.6% vs. 67.2%, P< 0.001) than non-progressors. Prevalence of diabetes mellitus and hypertension was higher among progressors to CKD than among non-progressors. Progressors to CKD showed higher baseline TAT and VAT than non-progressors. However, SAT was comparable between the groups.”

---

## [Decision Letter · Decision Letter 2]

8 Jan 2023

Association of intraabdominal fat with the risk of incident chronic kidney disease according to body mass index among Korean adults

PONE-D-22-12853R2

Dear Dr. Heo,

We’re pleased to inform you that your manuscript has been judged scientifically suitable for publication and will be formally accepted for publication once it meets all outstanding technical requirements.

Kind regards,

Paolo Magni

Academic Editor

PLOS ONE

Additional Editor Comments (optional):

All reviewer's comments have been correctly addressed.

Reviewers' comments:

Reviewer's Responses to Questions

**Comments to the Author**

1. If the authors have adequately addressed your comments raised in a previous round of review and you feel that this manuscript is now acceptable for publication, you may indicate that here to bypass the “Comments to the Author” section, enter your conflict of interest statement in the “Confidential to Editor” section, and submit your "Accept" recommendation.

Reviewer #3: All comments have been addressed

2. Is the manuscript technically sound, and do the data support the conclusions?

Reviewer #3: Yes

3. Has the statistical analysis been performed appropriately and rigorously? 

Reviewer #3: Yes

4. Have the authors made all data underlying the findings in their manuscript fully available?

Reviewer #3: No

5. Is the manuscript presented in an intelligible fashion and written in standard English?

Reviewer #3: Yes

6. Review Comments to the Author

Reviewer #3: (No Response)

7. PLOS authors have the option to publish the peer review history of their article (what does this mean?). If published, this will include your full peer review and any attached files.

Reviewer #3: No

---

## [Editor Report · Acceptance letter]

27 Jan 2023

PONE-D-22-12853R2 

Association of intraabdominal fat with the risk of incident chronic kidney disease according to body mass index among Korean adults 

Dear Dr. Heo:

I'm pleased to inform you that your manuscript has been deemed suitable for publication in PLOS ONE. Congratulations! Your manuscript is now with our production department. 

Kind regards, 

on behalf of

Prof. Paolo Magni 

Academic Editor

PLOS ONE